# An Ultra-Sensitive Technique: Using *Pv*-mtCOX1 qPCR to Detect Early Recurrences of *Plasmodium vivax* in Patients in the Brazilian Amazon

**DOI:** 10.3390/pathogens10010019

**Published:** 2020-12-30

**Authors:** Laila R. A. Barbosa, Emanuelle L. da Silva, Anne C. G. de Almeida, Yanka E. A. R. Salazar, André M. Siqueira, Maria das Graças Costa Alecrim, José Luiz Fernandes Vieira, Quique Bassat, Marcus V. G. de Lacerda, Wuelton M. Monteiro, Gisely C. Melo

**Affiliations:** 1Fundação de Medicina Tropical Dr. Heitor Vieira Dourado, Instituto de Pesquisa Clínica Carlos Borborema, Manaus 69040-200, AM, Brazil; laila_rowena@hotmail.com (L.R.A.B.); emanuellelira96@gmail.com (E.L.d.S.); anne.almeida.gb@gmail.com (A.C.G.d.A.); yankasalazar97@gmail.com (Y.E.A.R.S.); galecrim@fmt.am.gov.br (M.d.G.C.A.); marcus.lacerda@fiocruz.br (M.V.G.d.L.); wmonteiro@uea.edu.br (W.M.M.); 2Programa de Pós-Graduação em Medicina Tropical, Universidade do Estado do Amazonas, Manaus 69040-200, AM, Brazil; 3UNINILTONLINS—Universidade Nilton Lins, Unicenter, Manaus 69058-030, AM, Brazil; 4FAMETRO—Faculdade Metropolitana de Manaus, Campus Central, Av. Constantino Nery, Chapada, Manaus 69050-000, AM, Brazil; 5Fiocruz-Manguinhos—Fundação Oswaldo Cruz, Instituto de Pesquisa Clínica Evandro Chagas, Rio de Janeiro 21040-900, RJ, Brazil; andre.siqueira@ini.fiocruz.br; 6Instituto de Ciências da Saúde, Universidade Federal do Pará, Belém 66010-010, PA, Brazil; jvieira@ufpa.br; 7ISGlobal, Hospital Clínic, Universitat de Barcelona, 08036 Barcelona, Spain; quique.bassat@isglobal.org; 8Centro de Investigação em Saúde de Manhiça (CISM), Fundação Clínic per la Recerca Biomédica, Maputo 1929, Mozambique; 9Catalan Institution for Research and Advanced Studies (ICREA), Campus Clínic, Pg. Lluís Companys 23, 08010 Barcelona, Spain; 10Pediatric Infectious Diseases Unit, Pediatrics Department, Hospital Sant Joan de Déu, University of Barcelona, 1867 Barcelona, Spain; 11Consorcio de Investigación Biomédica en Red de Epidemiología y Salud Pública (CIBERESP), 28029 Madrid, Spain; 12Instituto Leônidas & Maria Deane, ILMD-Fiocruz-Fundação Oswaldo Cruz, Manaus 69057-070, AM, Brazil

**Keywords:** malaria diagnosis, submicroscopic infection, chloroquine resistance, mitochondrial DNA

## Abstract

Background: Early recurrence of *Plasmodium vivax* is a challenge for malaria control in the field, particularly because this species is associated with lower parasitemia, which hinders diagnosis and monitoring through blood smear testing. Early recurrences, defined as the persistence of parasites in the peripheral blood despite adequate drug dosages, may arise from resistance to chloroquine. The objective of the study was to estimate early recurrence of *P. vivax* in the Brazilian Amazon by using a highly-sensitive detection method, in this case, PCR. Methods: An ultra-sensitive qPCR that targeted mitochondrial DNA was used to compare a standard qPCR that targeted 18S rDNA to detect early recurrence of *P. vivax* in very low densities in samples from patients treated with chloroquine. Results: Out of a total of 312 cases, 29 samples (9.3%) were characterized as recurrences, from which 3.2% (10/312) were only detected through ultra-sensitive qPCR testing. Conclusions: Studies that report the detection of *P. vivax* early recurrences using light microscopy may severely underestimate their true incidence.

## 1. Introduction

*Plasmodium vivax* is estimated to be responsible for 7.5 million clinical cases of malaria worldwide in 2018, and 75% of this total occurs in the region of the Americas [1]. Chloroquine (CQ) in combination with primaquine (PQ) is the first-line treatment for *P. vivax* malaria in most endemic countries, but CQ efficacy is now under threat from the emergence and spread of CQ resistant species of *P. vivax,* the occurrence of which was first reported in Papua New Guinea in 1989 [2]. In areas where high-grade CQ-resistant *P. vivax* is prevalent, CQ-resistance is an important contributing factor to severe *P. vivax* malaria [3]. *P. vivax* resistance to CQ has been reported in the Amazon region in Brazil, with prevalence ranging from 4.4% to 11.5% [4,5,6,7].

In the absence of effective anti-hypnozoite treatment with 8-aminoquinoline, monitoring of CQ-resistant *P. vivax* malaria represents a special challenge due to the ability of a new infection to relapse from dormant liver stages over weeks to years. CQ-resistance is defined as the persistence or reappearance of parasites at plasma levels of CQ and its metabolite, desethylchloroquine (DCQ), above 100 ng/dL within the first 28 days (D28) after standard therapy with CQ [2,5,8]. Recurrent parasitemia that occurs after D28 is generally considered a relapse or new infection [8]. Higher CQ efficacy in preventing late recurrence compared with drugs with shorter half-lives, such as artemether–lumefantrine, has been observed in areas of low prevalence of CQ-resistance [9]. 

The usual methods for measuring CQ efficacy are prone to underestimating the true early recurrence rates, which is a fundamental barrier for adequate control and longer-term strategies targeting regional elimination. The aim of this study was to accurately estimate early recurrence of *P. vivax* in the Brazilian Amazon using a highly sensitive qPCR method. 

## 2. Results

During the period of the study, 450 patients treated with CQ were enrolled. However, only 312 samples were available on D28 for DNA extraction and qPCR testing. Thirty-four (10.9%) samples were positive by *Pv*-mtCOX1 qPCR, although 5 samples were excluded from the final analysis because CQ/DCQ concentrations were less than 100 ng/dL (Figure 1).

### 2.1. Pv-mtCOX1 qPCR, Pv 18S rRNA qPCR and Thick Blood Smear Agreement Results

Clinical and laboratory characteristics of patients included in this study are presented in Table 1. Of the 312 samples, 29 (9.3%) were positive for *Pv*-mtCOX1 qPCR, 14 (4.5%) after analysis using *Pv* 18S rRNA qPCR and only 5 (1.6%) were positive via thick blood smear (TBS). *Pv*-mtCOX1 qPCR was consistent with the positivity of the samples from the other two tests.

Among the 29 samples that were positive by *Pv*-mtCOX1 qPCR on D28, 11 (37.9%) of these patients were positive under TBS microscopy between day 30 and day 38 of follow-up, while 4 (13.8%) were positive on day 42 and 2 (1.4%) after 95 days. Agreement between positive diagnoses between the *Pv* 18S rRNA qPCR test and TBS microscopy was seen for only 37.5% of the samples, with a kappa coefficient of 0.51. Between *Pv*-mtCOX1 qPCR and TBS, co-positivity was 17.2%, with a kappa value of 0.27. When the two molecular tests were compared, co-positivity was 51.7%, with a kappa value of 0.63.

### 2.2. Parasitological Quantifications

Regarding quantifications, the average obtained for the TBS was 1710.6 parasites/µL blood, and averages using *Pv* 18S rRNA qPCR were 561.9 DNA copies/μL and 3188.9 DNA copies/μL using *Pv*-mtCOX1 qPCR. The mean quantification obtained after using *Pv*-mtCOX1 qPCR was significantly higher than when using *Pv* 18S rRNA qPCR (Figure 2a,b) (*p* < 0.00010). Correlation showed that quantifications of the two qPCRs tend to increase together (Pearson correlation coefficient = 0.387, *p* < 0.00, Figure 3).

### 2.3. Risk Factors

Univariate and multivariate (logistic regression) analyses and odds ratio were performed to determine certain potential risk factors, including being of the male gender, presence of gametocytes at microscopy on D0, asexual parasitemia in more than 5000 parasites and anemia on D0. Only the presence of gametocytes was found to be an independent risk factor, with an odds ratio (OR) of 3.98 in univariate analyses and 3.64 in multivariate analyses (*p* < 0.05 in both analyses, Table 2).

### 2.4. Symptomatic Recurrences 

Information obtained via the SIVEP-Malaria platform (the official Brazilian Malaria Epidemiological Surveillance Information System) indicated that 30.1% (94/312) of individuals were confirmed to have symptomatic recurrences after the study follow-up on D28. Furthermore, 17 out of 29 individuals were positive on D28 according to testing using *Pv*-mtCOX1 qPCR (Appendix A). The Kaplan–Meier method showed that the group that was positive on D28 when tested with *Pv*-mtCOX1 qPCR had recurrences earlier than the control group, with *p* < 0.01 (Figure 4). When analyzing clearance time by TBS soon after treatment initiation, no difference was observed between these groups (*p* > 0.05). 

## 3. Discussion

In this study, we compared different diagnostic methods for detecting early recurrence. *Pv*-mtCOX1 qPCR increased our ability to detect early recurrence. The quantification was superior and the samples with lower copy number were not detected through *Pv* 18S rRNA qPCR. These data confirm the high sensibility of the mitochondrial target gene, as shown previously in other studies [11,12,13,14,15] that reveal that it is present in at least 20 copies per cell [16].

These techniques are thousands of times more sensitive than rapid diagnostic tests (RDTs) and microscopy, and tens-to-hundreds of times more sensitive than standard PCR [17,18]. Gruenberg et al. [13] demonstrated the first target for use in an ultra-sensitive qPCR screening for *P. vivax*, namely qPCR *Pv*-mtCOX1. 

As noted in this study, other studies have also found that *Pv*MtCOX1 was more sensitive in the detection of *P. vivax* [11,14]. Hofman et al. [12] used *Pv*-mtCOX1 qPCR and obtained 62% positivity, while rRNA 18S only revealed 52% positivity in asymptomatic individuals from Papua New Guinea. Gruenberg et al. [14] observed that the use of *Pv*-mtCOX1 led to a significant increase in positivity in the range of 5.1%, 6.4% and 11.5% in Thailand, Brazil and Papua New Guinea, respectively. 

Moreover, yearly detection using *Pv*-mtCOX1 qPCR could predict the recurrence of symptoms in these patients, well before the patient is symptomatic and seeks health care. Sexual stages appear early in the course of *vivax* infection and in parallel with asexual parasitemia [19], therefore, asymptomatic infections are able to transmit to mosquitos due the presence of gametocytes even when there is low parasitemia [20,21]. Possibly these patients have been transmitting to mosquitos until the parasitemia grows sufficiently to be diagnosed by TBS and then be treated again. Due to the presence of asymptomatic carriers of Plasmodium being more frequent and probably serving as reservoirs for mosquitoes, the application of molecular techniques for diagnosis has been of fundamental importance [22].

In contrast to our findings, Lin et al. [23] did not find the same association between microscopic gametocyte carriage and parasite drug resistance, and that gametocytemia was more common in those reporting antimalarial use within the past year. 

However, this study had some limitations. It was not possible to differentiate between sexual and asexual forms of *P. vivax*, though we did perform clinical monitoring of the participants for a period of one year and, as a result, it was shown that malaria returned. In most cases, participants were positive using Pv-mtCOX1 and the recurrence time was shorter. Moreover, studies in Thailand and Indonesia demonstrated that without treatment *P. vivax* gametocytes are reported to persist in the peripheral circulation for a maximum of three days, and that the relationship between asexual and sexual stage parasitemia does not differ substantially between initial and recurrent infections [24]. Sexual stages appear early in the course of vivax malaria infection and increase jointly with the asexual parasitemia [19]. Another limitation was that data regarding recurrence episodes were accessed on the SIVEP-Malaria platform (Malaria Epidemiological Surveillance System), which is susceptible to underreporting issues.

## 4. Material and Methods

### 4.1. Ethics Statement, Study Site and Selection of Patients

The study was approved by the Ethics Review Board at Fundação de Medicina Tropical Doutor Heitor Vieira Dourado (FMT-HVD) (approval number: 1074306/2015). All participants signed an informed consent form. The study was performed at Fundação de Medicina Tropical Doutor Heitor Vieira Dourado (FMT-HVD), Manaus, AM and included patients from 2011 to 2017. 

The study included patients with *P. vivax* malaria of both genders, aged 6 months to 60 years, weight greater than 5 kg, with blood parasite density from 250 to 100,000 parasites/mL and axillary temperature ≥37.5 °C or history of fever in the last 48 h. Use of antimalarials in the previous 30 days, impossibility of being followed up for 42 days and clinical complications were considered exclusion criteria. Patients received treatment with 25 mg/kg of CQ phosphate over a period of 3 days (10 mg/kg on day 0 and 7.5 mg/kg on days 1 and 2) either associated or not with PQ 0.5 mg/kg/day for the following 7 days [25]. Patients were evaluated on days 0, 1, 2, 3, 7, 14, 28 and 42, as well as at any time during the follow-up period if they felt ill. 

### 4.2. P. vivax Malaria Diagnosis by Thick Smears

Thick blood smear (TBS) was performed using the Walker technique and performed by experienced microscopists [26]. Parasite densities were calculated by counting the number of parasites per 200 leukocytes, and the number of parasites/µL per patient was also determined.

### 4.3. Detection of Plasmodium spp. By qPCR

DNA extraction was performed using the PureLink^®^ Genomic DNA kit (Thermo Fisher Scientific, Waltham, MA, USA) following the manufacturer’s specifications. All DNA samples were subjected to *Pv*-mtCOX1 qPCR to detect *Plasmodium vivax* by targeting a conserved region of the *cytochrome oxidase 1* gene [15]. For quantification of *Pv*-mtCOX1 copy numbers, in each experiment these were quantified using a standard curve generated from three dilutions of plasmids containing the respective targeted region in triplicate (10^2^, 10^4^ and 10^6^ copies/μL). DNA was amplified using a 7500 fast Real-Time PCR System^®^ (Thermo Fisher Scientific), with the use of the primers and Taqman fluorescence probes described in Appendix A. Cycling parameters for PCR were an initial denaturation step at 95 °C for 10 min, 45 cycles of 15 s at 95 °C and 1 min at 60 °C. Each DNA samples was assayed in triplicate. All biology molecular procedures were performed at FMT-HVD.

*P. vivax* early recurrence was defined if qPCR was positive until D28. With CQ/DCQ plasma concentrations higher than 100 ng/dL, these patients were likely to have CQ-resistant parasites. In addition, individual data were collected from the Brazilian Malaria Epidemiological Surveillance System (SIVEP-Malaria) to verify their recurrence after the study follow-up.

### 4.4. CQ and DCQ Level 

CQ and DCQ plasmatic levels were determined only in case of parasitological failure. Three aliquots of 100 µL of whole blood samples were spotted on filter paper for analysis by high performance liquid chromatography (HPLC), as previous described [26,27]. The sum of CQ and DCQ plasmatic levels was used to classify parasites as being an early recurrence based on the 100 ng/mL cutoff level [28,29,30]. 

### 4.5. Statistical Analyses

Co-positivity and co-negativity between the tests and the agreement evaluation were calculated by the Kappa coefficient using *Pv*-mtCOX1 qPCR as the standard method. Univariate and multivariate analyses were performed to identify factors associated to early recurrence. The difference between the averages of parasite density was estimated by the non-parametric Mann–Whitney test, and all tests considered a 95% confidence interval. Analyses and graphs were generated by the software Statistical Package for the Social Sciences (IBM_SPSS) v.21 and GraphPad Prism 7 Software. Statistical analyses were performed at FMT-HVD.

## 5. Conclusions

We conclude that *Pv*-mtCOX1 qPCR is an ultrasensitive instrument for the diagnosis of early recurrence samples when compared to the other methods tested. The results obtained show the existence of sub-notification when tests are done using TBS and highlight the need for development and implementation of more sensitive tests, though not only for diagnosis but also for follow-up of patients with vivax malaria in order to favor more assertive decisions aimed at the elimination of the disease.

## Figures and Tables

**Figure 1 pathogens-10-00019-f001:**
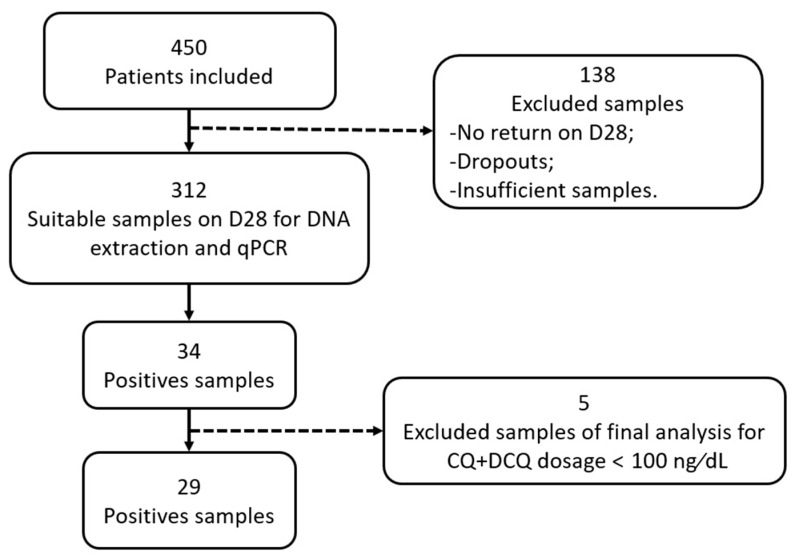
Flowchart of processed samples.

**Figure 2 pathogens-10-00019-f002:**
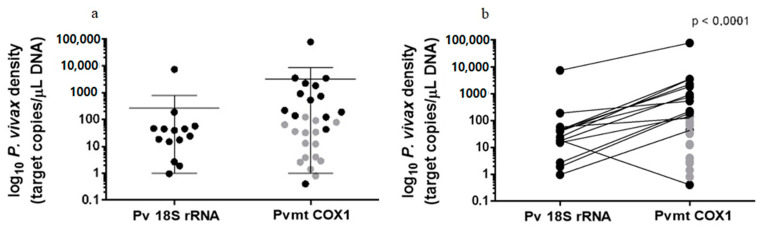
Copy numbers of each target per sample. (**a**) Each dot represents one sample; gray indicates samples detected only by *Pv*-mtCOX1 qPCR. (**b**) Lines linking the same samples in both targets show the different quantifications between them.

**Figure 3 pathogens-10-00019-f003:**
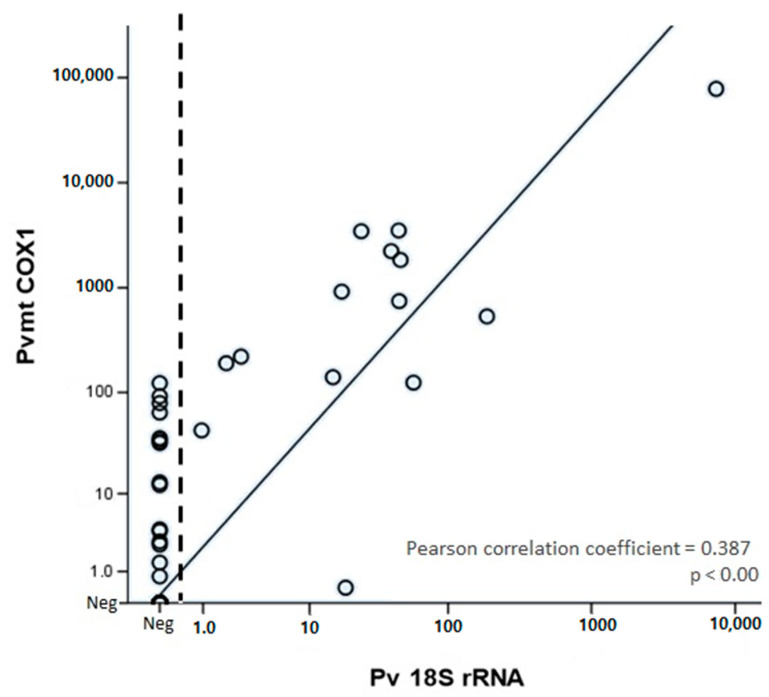
Correlation of log_10_ template copy numbers detected by *Pv*-mtCOX1 and *Pv* 18S rDNA qPCR.

**Figure 4 pathogens-10-00019-f004:**
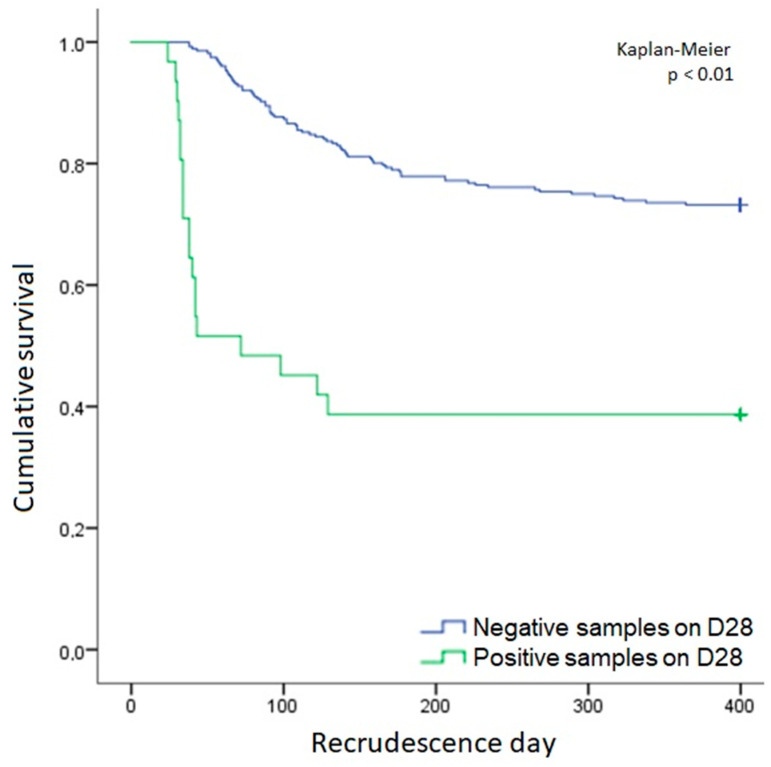
Survival function calculated by the Kaplan–Meier method in relation to the day of patient’s recrudescence after the study follow-up period, showing the curves of the negative and positive samples for *Pv*-mtCOX1 qPCR on day 28.

**Table 1 pathogens-10-00019-t001:** Demographic and clinical data of individuals involved in this study.

Parameter	Value
Sample size	307
Mean age, years (SD)	39.4 (14.3)
No. of males (%)	215 (70)
Parasite geometric mean D0 per mm^3^ (SD)	3781.6 (4109.2)
No. with presence of gametocytes D0 under microscopy (%)	259 (84.4)
Mean hemoglobin in D0, g/dL (SD) *^a^*	13.4 (1.7)
Mean hemoglobin in D28, g/dL (SD) *^a^*	13.6 (1.1)
No. with anemia (%) *^a^*^,*b*^	57 (27.5)

*^a^* Only 207 samples had data on hemoglobin. ^*b*^ According to World Health Organization (WHO) criteria (2001) [10].

**Table 2 pathogens-10-00019-t002:** Univariate and multivariate (logistic regression) analyses of factors associated in D0 to early recurrence in patients with uncomplicated *Plasmodium vivax* malaria under supervised treatment with chloroquine plus primaquine.

Factor	Univariate Analysis				Multivariate Analysis			
	Odds Ratio	95% Confidence Interval		*p*-Value	Adjusted Odds Ratio	95% Confidence Interval		*p*-Value
		Lower	Upper			Lower	Upper	
Male gender	0.72	0.30	1.76	0.471	0.91	0.29	2.84	0.872
Presence of gametocytes under microscopy	3.98	1.51	9.54	**0.001**	3.64	1.10	12.08	**0.035**
Asexual parasites > 5000/mm^3^	2.14	0.72	6.37	0.161	1.03	0.26	4.03	0.968
Anemia D0 *^a^*^,*b*^	1.65	0.42	5.76	0.497	1.51	0.40	5.73	0.541

Bold type indicates significance. *^a^* Only 207 samples had data on hemoglobin. *^b^* According to WHO criteria (2001) [10].

## Data Availability

The data that support the findings of this study are available from the corresponding author (GCM), upon reasonable request, due to ethical restrictions.

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
