# Peer review of "An Ultra-Sensitive Technique: Using Pv-mtCOX1 qPCR to Detect Early Recurrences of Plasmodium vivax in Patients in the Brazilian Amazon"

_pathogens, 2020, doi:10.3390/pathogens10010019_

Round 1

Reviewer 1 Report

The paper is well written and addresses a topic of great impact. The necessity of diagnosis of very low parasitaemias is common for differenet Plasmodium species, mostly for Plasmodium falciparum and Plasmodium vivax both for studying and monitoring parasites transmission and to study drug resistant in endemic populations. 

The comparison of two different qPCR methods, also with microscopy, for low density parasite identification open new perspectives on the field research and personally I'm very interested in the possibility to explore mithocondrial genes in qPCR for different Plasmodium species. 

Pictures and tables are simple and explanatory, the results are complete and the population characterization is well reported. 

I reccomend the authors to improve the methods section with more details on qPCR programs, steps, temperatures and controls for standard curves used in the different conditions, in order to allow the simple material preparation and the replication of the experiments. 

The paper gives to those who read it a complete view of both the initial purpose and the obtained results, for these reasons I reccomend the acceptance after the improvement in the methods section.

Author Response

Reviewer #1: Attached please find the final version of the manuscript entitle: “An ultra-sensitive technique: using Pv-mtCox1 qPCR to detect early recurrences of Plasmodium vivax in patients in the Brazilian Amazon” by Barbosa and co-authors. We have also included the answers and clarifications required by the reviewers. The paper is well written and addresses a topic of great impact. The necessity of diagnosis of very low parasitaemias is common for differenet Plasmodium species, mostly for Plasmodium falciparum and Plasmodium vivax both for studying and monitoring parasites transmission and to study drug resistant in endemic populations. The comparison of two different qPCR methods, also with microscopy, for low density parasite identification open new perspectives on the field research and personally I'm very interested in the possibility to explore mithocondrial genes in qPCR for different Plasmodium species. Pictures and tables are simple and explanatory, the results are complete and the population characterization is well reported. 

We thank this reviewer for insightful comments, which have improved significantly the quality of this manuscript, following the specific comments below.

1. I reccomend the authors to improve the methods section with more details on qPCR programs, steps, temperatures and controls for standard curves used in the different conditions, in order to allow the simple material preparation and the replication of the experiments. The paper gives to those who read it a complete view of both the initial purpose and the obtained results, for these reasons I reccomend the acceptance after the improvement in the methods section.

All DNA samples were subjected to PvMtCox Taqman qPCR to detect Plasmodium vivax by targeting a conserved region of the cytochrome oxidase 1 gene (Gruenberg et al., 2020). For quantification of PvMtCox copy numbers, in each experiment were quantified using a standard curve generated from three dilutions of plasmids containing the respective region targeted were included in triplicates (102, 104 and 106 copies/μl). DNA was amplified in 7500 fast Real-Time PCR System®, using the primers and Taqman fluorescence probes describe in Additional file 1.  Cycling parameters for PCR were an initial denaturation step at 95 °C for 10 minutes, 45 cycles of 15 seconds at 95 °C and 1 minute at 60 °C. Each DNA samples was assayed in triplicated. Information has been added to the manuscript in the lines166-175.

Reviewer 2 Report

The manuscript presented by Barbosa and colleagues aims to estimate early recurrence of Plasmodium vivid in the Brazilian Amazon using a highly sensitive qPCR method for detection.

The data presented by the authors are important in the context of human malaria, however some corrections need to be considered, namely:

1. I think that figure 2 is unnecessary in the manuscript because the data presented can be presented in the description of the results/ discussion;

2. I think the discussion could be better explored by the authors. The proposal of the manuscript is not in the short communication format;

3. Section "Material and methods" - I think that items 4.1, 4.2, 4.3 could be grouped into just one item;

4. Lines 170-173 - I think that the methodology presented in the section should be better described and references presented.

Author Response

Reviewer #2: The manuscript presented by Barbosa and colleagues aims to estimate early recurrence of Plasmodium vivid in the Brazilian Amazon using a highly sensitive qPCR method for detection. The data presented by the authors are important in the context of human malaria, however some corrections need to be considered, namely:

1. I think that figure 2 is unnecessary in the manuscript because the data presented can be presented in the description of the results/ discussion;

Clarification was made accordingly.

2. I think the discussion could be better explored by the authors. The proposal of the manuscript is not in the short communication format;

Change was made accordingly.

3. Section "Material and methods" - I think that items 4.1, 4.2, 4.3 could be grouped into just one item;

Change was made accordingly.

4. Lines 170-173 - I think that the methodology presented in the section should be better described and references presented.

Three aliquots of 100µL of whole blood samples were spotted in filter paper for analysis by high performance liquid chromatography (HPLC), as previous described (Dua et al., 1999; Patchen et al., 1983). The sum of CQ and DCQ blood levels was used to classify parasites as early recurrence based on the 100-ng/mL cutoff level (WHO, 2009; Price et al., 2014). Information has been added to the manuscript in the lines 181-185.

This manuscript is a resubmission of an earlier submission. The following is a list of the peer review reports and author responses from that submission.